# Complex Permittivity and Microwave Absorption Properties of OPEFB Fiber–Polycaprolactone Composites Filled with Recycled Hematite (α-Fe_2_O_3_) Nanoparticles

**DOI:** 10.3390/polym11050918

**Published:** 2019-05-24

**Authors:** Ebenezer Ekow Mensah, Zulkifly Abbas, Raba’ah Syahidah Azis, Nor Azowa Ibrahim, Ahmad Mamoun Khamis

**Affiliations:** 1Department of Physics, Faculty of Science, Universiti Putra Malaysia, 43400 Serdang, Selangor, Malaysia; ebemensek@yahoo.co.uk (E.E.M.); rabaah@upm.edu.my (R.S.A.); akhameis@yahoo.com (A.M.K.); 2Institute of Advanced Materials, Universiti Putra Malaysia, 43400 UPM Serdang, Selangor, Malaysia; 3Department of Chemistry, Faculty of Science, Universiti Putra Malaysia, 43400 Serdang, Selangor, Malaysia; norazowa@upm.edu.my

**Keywords:** recycled hematite, mill scale, mechanical alloying, complex permittivity, reflection loss

## Abstract

Recycled hematite (α-Fe_2_O_3_) nanoparticles with enhanced complex permittivity properties have been incorporated as a filler in a polycaprolactone (PCL) matrix reinforced with oil palm empty fruit bunch (OPEFB) fiber for microwave absorption applications. The complex permittivity values were improved by reducing the particle sizes to the nano scale via high-energy ball milling for 12 h. A total of 5–20 wt.% recycled α-Fe_2_O_3_/OPEFB/PCL nanocomposites were examined for their complex permittivity and microwave absorption properties via the open ended coaxial (OEC) technique and the transmission/reflection line measurement using a microstrip connected to a two-port vector network analyzer. The microstructural analysis of the samples included X-ray diffraction (XRD), high-resolution transmission electron microscopy (HRTEM) and Fourier transform infrared spectroscopy (FTIR). At 1 GHz, the real (*ε′*) and imaginary (*ε″*) parts of complex permittivity of recycled α-Fe_2_O_3_ particles, respectively, increased from 7.88 to 12.75 and 0.14 to 0.40 when the particle size was reduced from 1.73 μm to 16.2 nm. A minimum reflection loss of −24.2 dB was achieved by the 20 wt.% nanocomposite at 2.4 GHz. Recycled α-Fe_2_O_3_ nanoparticles are effective fillers for microwave absorbing polymer-based composites in 1–4 GHz range applications.

## 1. Introduction

The sharp growth in the electronics industry spanning the microwave frequency range and its associated electromagnetic interference (EMI) problem has led to intensified research into the development of microwave absorbing materials to reduce the impact of EMI. In recent years, ferrite–polymer composites based on spinel or hexagonal ferrites and non-degradable polymer matrices such as epoxy resin [1,2] polyvinylchloride [3] and polyaniline [4,5], have been studied. For an effective microwave absorption, materials having low density, strong mechanical properties, thinness, strong absorption and low-cost are preferred for several applications.

Ferrites have attracted a lot of attention in microwave absorber development in view of their excellent electrical and magnetic properties, and are often synthesized through multi-staged chemical techniques such as sol–gel [6], co-precipitation [1], solid-state [2] and hydrothermal [7] reactions. Among the less extensively used ferrites for microwave absorption applications is hematite (α-Fe_2_O_3_), a ferrite (corundum-type oxide) which is stable under ambient conditions with unique electrical and magnetic properties. Recently, α-Fe_2_O_3_ was successfully synthesized from industrial mill scale waste material [8,9] through a low-cost technique. Recycled α-Fe_2_O_3_ is cheap, reduces environmental pollution and its generally low dielectric loss properties [10] can be enhanced by reducing the particles to the nanosize for use in polymer composites for microwave absorption applications.

Additionally, the dielectric loss properties of the recycled α-Fe_2_O_3_/polymer nanocomposites could further be enhanced by reinforcing with carbon bio-based materials for higher microwave absorbing capacities. Carbon bio-based materials from agricultural waste can serve as a low-cost and eco-friendly alternative to the often-used carbon nanotubes (CNT) and graphene. Abdalhadi et al. [11] reported in a recent study that compressed oil palm empty fruit bunch (OPEFB) fiber of 100 μm grain size possessed a high loss factor in the 1–4 GHz range. Moreover, OPEFB is an abundant waste material generated by the oil palm industries and the biodegradable fiber has low density, good mechanical and thermal properties. Reinforcing materials with OPEFB fiber could reduce cost, improve dielectric loss properties, enhance thermal stability and improve rigidity. OPEFB fiber and ferrites are easily blended with polymers such as polycaprolactone (PCL), as demonstrated in a study by Ahmad et al. [12].

PCL is a fully biodegradable polymer, easy to blend, lightweight, non-toxic and has good dielectric characteristics. Biodegradable composites reduce the effect on the environment when they are no longer in use since they can be broken down by micro-organisms. By combining a biodegradable polymer with recycled α-Fe_2_O_3_ and carbon bio-based material, a low-cost, light weight magneto-dielectric absorber with increased biodegradability and attractive microwave absorption properties could be fabricated, as reported in a similar work [13].

In this study, α-Fe_2_O_3_ was recycled from mill scale and the particle size reduced to the nanosize through high-energy ball milling in order to enhance the dielectric loss properties. A total of 5–20 wt.% recycled α-Fe_2_O_3_ nanofiller loadings were incorporated into OPEFB fiber/PCL matrix to fabricate nanocomposites using the melt–blend technique followed by hot molding. The samples were then evaluated for their complex permittivity and microwave reflection loss properties in the 1–4 GHz range.

## 2. Materials and Methods

### 2.1. Materials

The following materials were used for the preparation of the recycled α-Fe_2_O_3_ and the α-Fe_2_O_3_/OPEFB/PCL nanocomposites: Mill scale flakes (Perwaja Sdn. Bhd. Terengganu, Malaysia), Polycaprolactone (Sigma-Aldrich, St. Louis, MO, USA) 97.0% purity, density 1.146 g/cm^3^, Purified OPEFB (Dengkil, Malaysia) ground into 100 μm grain size.

### 2.2. Synthesis of α-Fe_2_O_3_ Nanoparticles from Mill Scale

The mill scale flakes were crushed into coarse particles and purified using magnetic and curie separation techniques [8,9]. The wustite (FeO) slurry produced was filtered, dried and oxidized at 600 °C for 6 h (holding time) in a Protherm furnace to obtain the recycled α-Fe_2_O_3_ which was then milled. The milling was performed at room temperature for 12 h using SPEX Sample Prep 8000D High-energy Ball Mill (SPEX SamplePrep LLC, Metuchen, NJ, USA) fitted with a 1425 rpm and 50 Hz motor operating at a clamp speed of 875 cycles/minute and a powder-to-ball ratio of 1:5. For every 50 min of milling, the steel vials were opened for 2 min so as to avoid transforming α-Fe_2_O_3_ to Fe_3_O_4_ [14].

### 2.3. Fabrication of Recycled α-Fe_2_O_3_/OPEFB/PCL Nanocomposites

The recycled α-Fe_2_O_3_/OPEFB/PCL nanocomposites were fabricated by mixing 5%, 10%, 15% and 20% mass percentages of recycled α-Fe_2_O_3_ nanoparticles with PCL and OPEFB fiber at a fixed mass ratio of 3:7 through the melt blend technique using Brabender Plastograph twin screw extruder (Model 815651, Brabender GmbH & Co. KG, Duisburg, Germany) at a temperature of 65 °C. The blended nanocomposites were then hot pressed at a pressure of 110 kg/cm² into 6 cm × 3.6 cm × 0.70 cm pellets for characterization. The synthesis process is as depicted in Figure 1.

### 2.4. Characterizations

#### 2.4.1. Structure and Composition

The phase composition and structure of the recycled α-Fe_2_O_3_ particles were analyzed using X-ray diffraction (XRD) on a fully automated Philips X-pert system (Model PW3040/60 MPD, Amsterdam, The Netherlands) with Cu-Kα radiation operating at a voltage of 40.0 kV, a current of 40.0 mA and at a wavelength of 1.5405 Å. The diffraction patterns were recorded in the 2θ range of 10–80° with scanning speed of 2 °/min. All data were subjected to the Rietveld analysis on PANalytic X’Pert Highscore Plus v3.0 software (PANalytical B.V., Almelo, The Netherlands). The samples were identified by comparing their diffraction peaks with the Inorganic Crystal Structure Database (ICSD).

The size, shape and arrangement of the particles were investigated using the JEM-2100F high resolution transmission electron microscope (HRTEM, JEOL, Tokyo, Japan). The HRTEM images were processed with the ImageJ software (Version 1.50i, NIH, University of Wisconsin, Madison, WI, USA, 2016) to obtain the particle size distribution. The dispersion of recycled α-Fe_2_O_3_ nanoparticles in the PCL/OPEFB matrix was evaluated with JEOL JSM-7600 Field Emission Scanning Electron Microscope (FESEM, JEOL, Tokyo, Japan). The specific surface areas of the α-Fe_2_O_3_ particles were determined by Brunauer–Emmett–Teller (BET) analysis through a Quantachrome Autosorb-1 pore size and surface area analyzer (Quantachrome Instruments, Boynton Beach, FL, USA). Fourier transform infrared spectroscopy (FTIR) analysis of the samples was carried out with the Perkin Elmer Spectrum100 FTIR Spectrometer (PerkinElmer, Inc., Waltham, MA, USA) within the wavelength range of 400–4000 cm^−1^.

#### 2.4.2. Complex Permittivity

The complex permittivity measurements were carried out at room temperature in the 1–4 GHz range using Agilent 85070B open ended coaxial (OEC) dielectric probe connected to the Agilent N5230A PNA-L Network Analyzer (Agilent Technologies, Santa Clara, CA, USA). This technique is appropriate for complex permittivity determination for liquids, semi-solids and solids having flat and smooth surfaces and with a gap-free contact between the probe surface and the materials. A standard one-port, short-air-water calibration was undertaken and a reference standard material (unfilled polytetrafluoroethylene) was characterized to confirm the accuracy of the calibration. The powdered samples were compressed into sample holders to a thickness of 6 mm. Measurements were taken by pressing the probe flat onto the surfaces of the samples while ensuring air-gap free contact throughout the process. The measurement set-up is as illustrated in Figure 2.

#### 2.4.3. Microwave Absorption

The reflection coefficient (S_11_) and transmission coefficient (S_21_) of the nanocomposites were measured using the transmission/reflection line technique based on RT duriod 5880 microstrip (length = 6.0 cm, width 5.0 cm, thickness 0.15 cm) connected to a two-port Anritsu MS 2024B VNA Master (Anritsu Corporation, Kanagawa, Japan). A standard calibration (Short, Open and Load) was performed to ensure accuracy of data acquisition. The measurements were carried out by placing the samples flat onto the surface of the microstrip while avoiding any air gap with the strip line. The variation in reflection loss (dB) with frequency in the 1–4 GHz range was then investigated.

## 3. Results and Discussion

### 3.1. Structure and Composition

The XRD patterns of the recycled α-Fe_2_O_3_ particles before and after 12 h of ball milling are shown in Figure 3. The diffractograms were compared with ICSD standard patterns and all the Bragg peaks of the particles were identified as single phase, with hexagonal (rhombohedral) structure of α-Fe_2_O_3_ belonging to the R–3c space group. These results are consistent with previously reported studies on α-Fe_2_O_3_ [15,16]. Moreover, the identicalness in crystal structure between the milled particles (ICSD: 98-009-4106) and the unmilled particles (ICSD: 98-002-2616) essentially suggests that the α-Fe_2_O_3_ did not transform to Fe_3_O_4_ during the high-energy ball milling since no other phases were identified. Additionally, the peaks of the milled particles broadened while the sharpness decreased, possibly due to the reduction in the crystallite sizes [17].

The average crystallite sizes were estimated from the Scherrer formula given by:(1)$$D=\frac{k\lambda }{B\cos\theta }$$
where *D* is the crystallite size, *B* is the Full Width at Half Maximum (FWHM) of the diffraction peaks in radians, *k =* 0.9, *Ɵ* is the peak position and *λ* = 1.5405 Å. As depicted in Table 1, the average crystallite sizes of the recycled α-Fe_2_O_3_ particles reduced from 106.2 to 11.1 nm after 12 h of high-energy ball milling.

The BET-specific surface area analysis of the particles was examined through the nitrogen adsorption/desorption method and the results are also shown in Table 1. The surface areas of the unmilled and milled particles were, respectively, found to be 0.202 and 13.159 m^2^/g, indicating an increase in specific surface area with reduced particle size after milling, which is attributed to the higher bulk density of the nanoparticles due to their decreased volume. The BET-specific surface area is an indication of the exposed surface area per mass unit of the particles and is a function of bulk density and particle size (diameter) as given by the equation [18];
(2)$$D_{BET}=\frac{6}{\rho \times SSA}$$
where $D_{BET}$ represents the particle size (diameter) and *ρ, SSA* are the bulk density and specific surface area respectively.

HRTEM micrographs and the size distributions of the recycled α-Fe_2_O_3_ particles are illustrated in Figure 4. It can be observed that initially the recycled α-Fe_2_O_3_ particles were loosely formed, randomly shaped and bulky. However, after 12 h of high-energy ball milling, nanoparticles with distinct aggregation and agglomeration were formed. The agglomeration could be due to higher particle–particle interaction arising from the increased specific surface area [19]. The nanoparticles were largely spherical and the sizes ranged from 10.3 to 24.45 nm with an average of 16.2 nm, consistent with the estimated crystallite sizes from the XRD analysis. The variations of the particle size with milling time for the recycled α-Fe_2_O_3_ particles are summarized in Table 1.

The surface morphologies of the recycled α-Fe_2_O_3_/OPEFB/PCL nanocomposites are presented in Figure 5. The FESEM micrographs show that there was a uniform dispersal of recycled α-Fe_2_O_3_ nanoparticles which appeared as spherical spots throughout the fractured surfaces. The dispersal increased as the recycled α-Fe_2_O_3_ nanofiller loadings increased in the nanocomposites. The dispersal of recycled α-Fe_2_O_3_ as discrete nanoparticles in the OPEFB-PCL matrix indicates that the nanoparticles were fully embedded in the nanocomposites to form a homogeneous mixture and provided interfacial bonding for the enhancement of the properties of the composites.

FTIR spectroscopy was performed to establish the functional groups present in recycled α-Fe_2_O_3_ nanoparticles, OPEFB fiber and PCL in order to identify the type of the interaction that exists in the nanocomposites. As illustrated in Figure 6, the spectrum for α-Fe_2_O_3_ shows characteristic peaks at 507 and 420 cm^−1^ which can be ascribed to Fe–O stretching/vibrational modes [20]. The pure PCL spectrum indicates characteristic peaks at 727, 1168, 1366, 1720 and 2940 cm^−1^ consistent with similar studies [12,21]. The spectrum for OPEFB portrays a characteristic absorption band at 3325 cm^−1^ which could be related to O–H stretching vibration [22]. Characteristic peaks were also found at 1635 cm^−1^ attributed to absorbed water by cellulose, 2919 cm^−1^ (C–H stretching vibration) and 1029 cm^−1^ (C–O stretching). It is evident that the spectra for α-Fe_2_O_3_/OPEFB/PCL nanocomposites comprise the characteristic peaks and bands of α-Fe_2_O_3_ nanofiller, PCL polymer and OPEFB fiber. The absence of any significantly new bands or changes in the peak positions in the spectra of the nanocomposites shows that there was no strong interaction between the recycled α-Fe_2_O_3_ nanofiller, PCL polymer and OPEFB fiber. The FTIR results, therefore, suggest that the mixture of recycled α-Fe_2_O_3_ nanoparticles, OPEFB fiber and PCL to form the nanocomposites was physical in nature.

### 3.2. Complex Permittivity

In order to investigate the dielectric properties of the recycled α-Fe_2_O_3_ particles, compacted samples were characterized for their real (*ε′*) and imaginary (*ε″*) parts of the relative complex permittivity (*ε* = ε′* – *jε″*). Compacting the samples decreased the void and eliminated air gaps likely to affect the results. The dielectric properties of recycled α-Fe_2_O_3_ particles were examined for reduced particle size dependency, and the variation with frequency is presented in Figure 7. Generally, *ε′* of the recycled α-Fe_2_O_3_ particles decreased with increase in frequency, a trend common with ferrites and explained based on the Maxwell–Wenger polarization model [23]. The *ε″* values, however, depicted an increasing profile with frequency, consistent with the OEC calibration profile of water in the studied frequency range. Additionally, both *ε′* and *ε″* values showed instabilities in their profiles due to impedance mismatch between the input impedance of the OEC probe and the surface impedance of the compacted samples arising from surface imperfections and voids. The 16.2 nm recycled α-Fe_2_O_3_ particles had higher *ε′* and *ε″* values than the 1.73 μm particles throughout the 1–4 GHz range. The *ε′* of the 16.2 nm particles varied from 12.75 to 12.03 while the *ε″* were between 0.40 and 0.54. For the 1.73 μm recycled α-Fe_2_O_3_ particles, the *ε′* ranged from 7.88 to 7.54 while the *ε″* values varied between 0.14 and 0.51. These results demonstrate that the complex permittivity of recycled α-Fe_2_O_3_ particles are significantly enhanced when the particle sizes are reduced to a nanosize.

The significantly higher permittivity values of the 16.2 nm recycled α-Fe_2_O_3_ particles can be related to the larger specific surface area and interfacial density resulting in the improvement in interfacial polarization. Additionally, ferrites and other 3-d metal oxide nanoparticles are susceptible to oxygen vacancy formation at the interfacial layers which can greatly influence the magnetic [24] and dielectric [25] properties. Moreover, the compactness of the nanoparticles enabled air-gap-free contact with constituent particles which led to the enhancement of interfacial polarization and therefore complex permittivity.

The effect of the 5–20 wt.% 16.2 nm recycled α-Fe_2_O_3_ nanofiller loadings on the dielectric permittivity of recycled α-Fe_2_O_3_/OPEFB/PCL nanocomposites was investigated, and the variation in *ε′* and *ε″* values in the 1–4 GHz range are presented in Figure 8. Clearly, the *ε′* and *ε″* of the nanocomposites increased with higher content of the recycled α-Fe_2_O_3_ nanofiller throughout the studied frequency range. Table 2 shows the *ε′* and *ε″* values at some specified frequencies and it can be observed that the *ε′* decreased with frequency while the *ε″* values increased with frequency for the 5, 10, 15 and 20 wt.% nanocomposites. As expected, the high complex permittivity values of the recycled α-Fe_2_O_3_ nanofiller and OPEFB fiber contributed to the nanocomposites exhibiting higher permittivity values with increased loadings, in agreement with the results of other inorganic/organic-polymer matrix composites [21]. Complex permittivity properties of a material largely depend on contributions of various forms of polarizations such as interfacial, atomic, orientation and electronic. However, the inclusion of OPEFB fiber in the PCL matrix expectedly enhanced the combined effect of orientation polarization and interfacial polarization [26] as well as the hopping exchange between O^2−^ and Fe^3+^ ions at the localized states [26,27] which caused the increase in the permittivity values.

### 3.3. Microwave Absorption

The microwave absorption properties of the recycled α-Fe_2_O_3_/OPEFB/PCL nanocomposites were deduced from the calculated reflection loss (*RL*) values using the measured reflection coefficient magnitudes (|*S*_11_|) obtained via the two-port VNA. The *RL* values were determined from the expression given by [28];
(3)$$RL=20\log\left|S_{11}\right|$$

The variation of *RL* with frequency for the nanocomposites are shown in Figure 9. Generally, the 5–20 wt.% α-Fe_2_O_3_ nanofiller loadings correlated with the decreasing *RL* values of the nanocomposites throughout the 1–4 GHz range. As indicated in Table 3, the most prominent band was located around 2.4 GHz where the minimum value reached −24.2 dB and the maximum −21.5 dB. All the *RL* values for the nanocomposites were also found to be less than −10 dB (90% absorption) with the 20 wt.% nanocomposites obtaining the lowest value. The improvement in the absorption capacity of the nanocomposites could be related to their high dielectric loss properties as a result of the combined effect of enhanced polarization attributed to the recycled α-Fe_2_O_3_ nanofiller and OPEFB fiber. These results demonstrate that the recycled α-Fe_2_O_3_/OPEFB/PCL nanocomposites are capable of significant microwave absorption and would serve as a cheaper and environmentally friendly alternative for applications within the investigated frequency range.

## 4. Conclusions

α-Fe_2_O_3_ was effectively recycled from mill scale waste material and the complex permittivity improved significantly by reducing the particle size through high-energy ball milling for 12 h. The imaginary part of complex permittivity (*ε″*), essential for microwave absorption properties, increased from 0.14 to 0.40 at 1 GHz when the particle size was reduced from 1.73 to 16.2 nm. The 16.2 nm recycled α-Fe_2_O_3_ nanoparticles were used as filler in PCL matrix reinforced with 100 μm OPEFB fiber to fabricate recycled α-Fe_2_O_3_/OPEFB/PCL nanocomposites which showed increased complex permittivity values and good absorption properties with increased nanofiller content. A minimum reflection loss of −24.2 dB was obtained for the 20 wt.% nanocomposite at 2.4 GHz. The recycled α-Fe_2_O_3_/OPEFB/PCL nanocomposites are cheap, easy to fabricate and are effective for microwave absorption applications in the 1–4 GHz range.

## Figures and Tables

**Figure 1 polymers-11-00918-f001:**
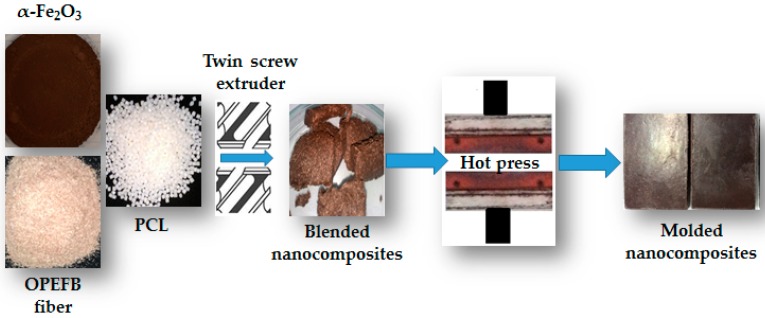
Nanocomposite fabrication. PCL, polycaprolactone.

**Figure 2 polymers-11-00918-f002:**
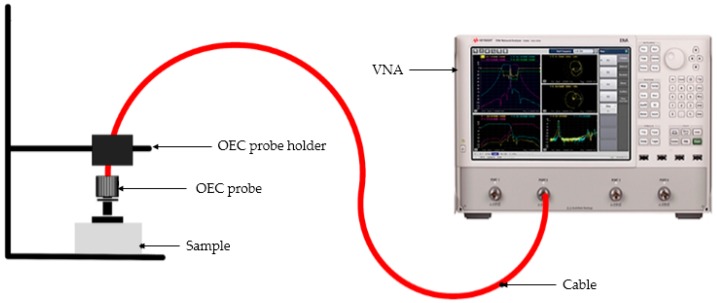
Complex permittivity measurement set-up. OEC, open ended coaxial.

**Figure 3 polymers-11-00918-f003:**
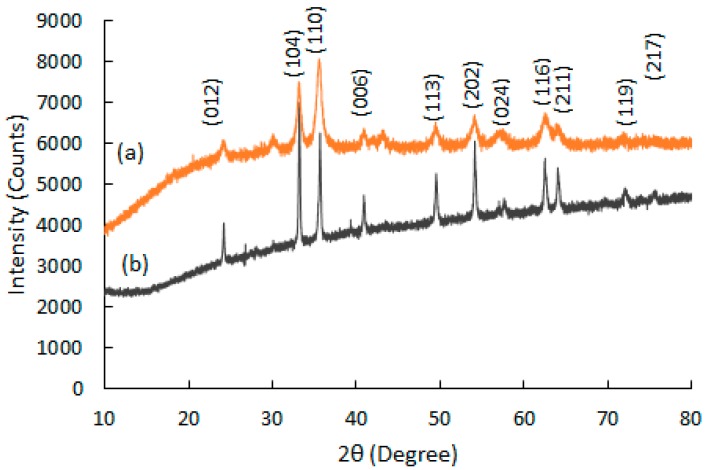
X-ray diffractograms for (a) α-Fe_2_O_3_ milled for 12 h (*R*_wp_ = 4.09%, Gof = 9.78); (b) unmilled α-Fe_2_O_3_ (*R*_wp_ = 2.43%, Gof = 2.27).

**Figure 4 polymers-11-00918-f004:**
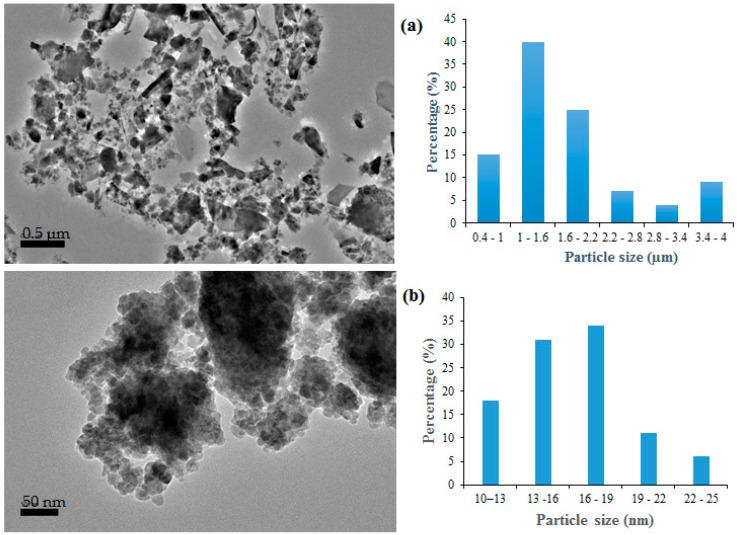
High-resolution transmission electron microscopy (HRTEM) micrographs and particle size distribution of α-Fe_2_O_3_ (**a**) unmilled (**b**) nanoparticles.

**Figure 5 polymers-11-00918-f005:**
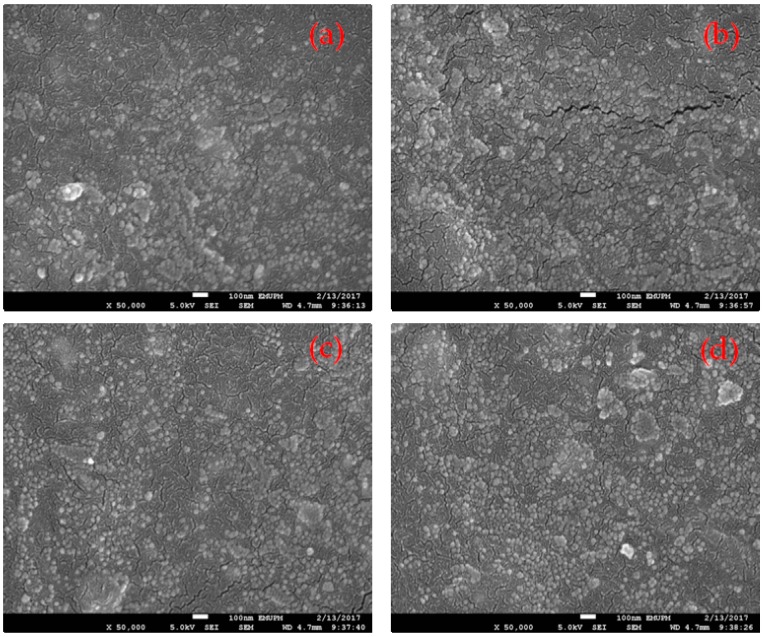
Field Emission Scanning Electron Microscope (FESEM) micrographs of recycled α-Fe_2_O_3_/OPEFB/PCL nanocomposites with (**a**) 5% (**b**) 10% (**c**) 15% (**d**) 20% recycled α-Fe_2_O_3_ nanofiller content. OPEFB, Oil palm empty fruit bunch fiber.

**Figure 6 polymers-11-00918-f006:**
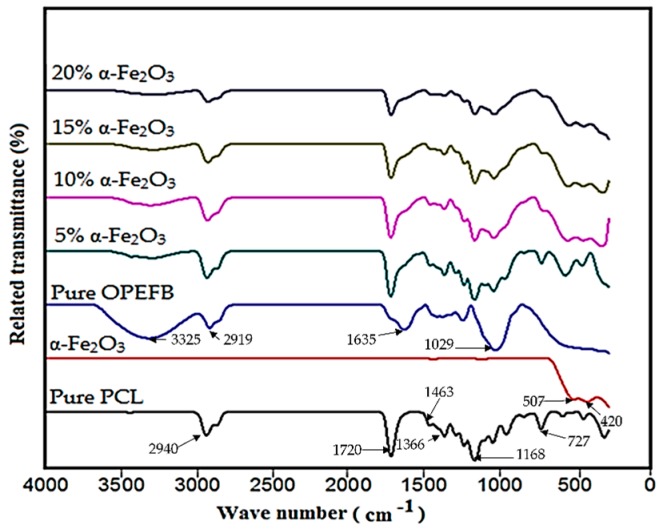
FTIR spectra of pure PCL, recycled α-Fe_2_O_3_, pure OPEFB fiber and α-Fe_2_O_3_/OPEFB/PCL nanocomposites.

**Figure 7 polymers-11-00918-f007:**
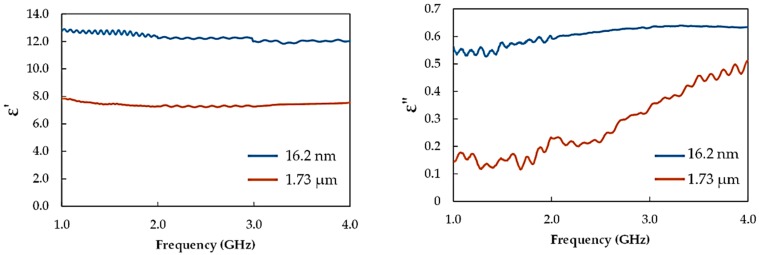
Complex permittivity of recycled α-Fe_2_O_3_ particles.

**Figure 8 polymers-11-00918-f008:**
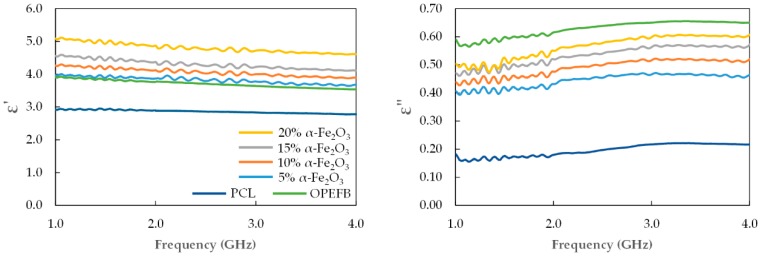
Complex permittivity of recycled α-Fe_2_O_3_/OPEFB/PCL nanocomposites.

**Figure 9 polymers-11-00918-f009:**
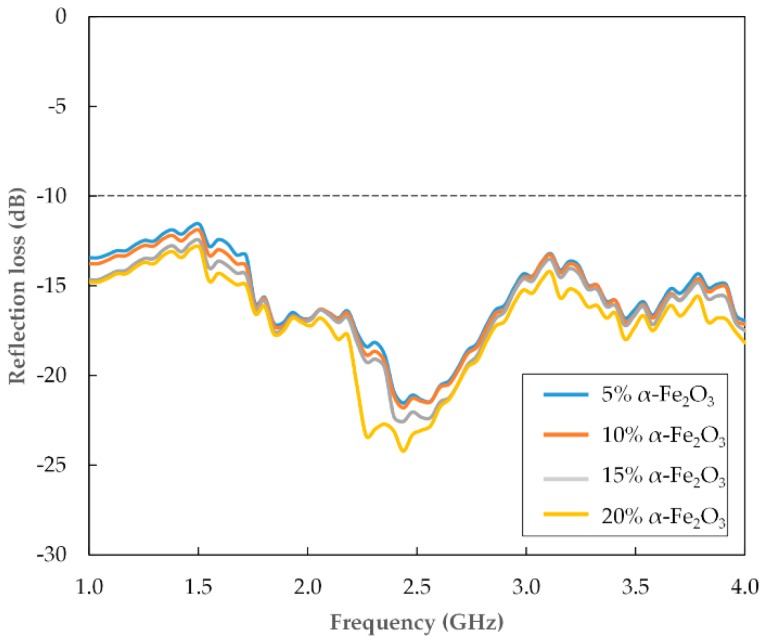
Variation in reflection loss with for the recycled α-Fe_2_O_3_/OPEFB/PCL nanocomposites.

**Table 1 polymers-11-00918-t001:** Crystallite size, particle size and specific surface area as a function of milling time.

α-Fe_2_O_3_ Sample	Crystallite Size (nm)	Particle Size	Specific Surface Area (m^2^/g)
Unmilled	106.2	1.73 (μm)	0.202
12 h milling	11.1	16.2 (nm)	13.159

**Table 2 polymers-11-00918-t002:** Variation in *ε′* and *ε″* values at specified frequencies.

α-Fe_2_O_3_ (wt.%)	1 GHz		2.4 GHz		4 GHz	
	*ε′*	*ε″*	*ε′*	*ε″*	*ε′*	*ε″*
5	3.97	0.40	3.88	0.45	3.67	0.46
10	4.26	0.43	4.10	0.49	3.89	0.52
15	4.56	0.47	4.32	0.53	4.11	0.57
20	5.08	0.50	4.82	0.57	4.60	0.60

**Table 3 polymers-11-00918-t003:** Variation in reflection loss (dB) for recycled α-Fe_2_O_3_/OPEFB/PCL nanocomposites.

α-Fe_2_O_3_ (wt.%)	1 GHz	2.4 GHz	4 GHz
5	−13.4	−21.5	−16.9
10	−13.7	−21.8	−17.1
15	−14.6	−22.5	−17.4
20	−14.7	−24.2	−18.1

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
