# Peer review of "Complex Permittivity and Microwave Absorption Properties of OPEFB Fiber–Polycaprolactone Composites Filled with Recycled Hematite (α-Fe2O3) Nanoparticles"

_polymers, 2019, doi:10.3390/polym11050918_

Round 1

Reviewer 1 Report

Referee Report

on paper “ Complex Permittivity and Microwave Absorption Properties of OPEFB Fiber – Polycaprolactone Composites Filled with Recycled Hematite (α – Fe2O3) Nanoparticles ” (polymers-479619) by authors Ebenezer Ekow Mensah, Zulkifly Abbas, Raba’ah Syahidah Azis, Nor Azowa Ibrahim and Ahmad Mamoun Khamis submitted to Polymers

This is interesting paper. It reports a complex permittivity and microwave absorption properties of recycled hematite α–Fe2O3 nanoparticles with enhanced complex permittivity incorporated as a filler in a polycaprolactone (PCL) matrix reinforced with oil palm empty fruit bunch (OPEFB). The complex permittivity values were improved by reducing the particle sizes to nano size via high energy ball milling for 12 hours. 5 to 20% wt. recycled α–Fe2O3/OPEFB/PCL nanocomposites were examined for their complex permittivity and microwave absorption properties. At 1 GHz, the real (ε') and imaginary (ε") parts of complex permittivity of recycled α–Fe2O3 particles respectively increased from 7.88 to 12.75 and 0.14 to 0.40 when the particle size was reduced from 1.73 μm to 16.2 nm. A maximum reflection loss of –24.2 dB was achieved by the 20% wt. nanocomposite at 2.4 GHz. Obtained experimental results are reliable without any doubts. A convincing interpretation has been given. However, a few points should be improved. I think that this paper can be published only after corresponding corrections :

1.    The choice of the research object is attractive for me. I fully agree with authors that “ Ferrites have attracted a lot of attention in microwave absorber development in view of their excellent electrical and magnetic properties …  ”. However, I some disagree with the authors that “  … although they are often synthesized through generally expensive, multi – staged and complicated chemical processes. ”. Good ferromagnetic nature for this type of materials, i.e. large total magnetic moment at operating temperature – ambient temperature – is also very important. Substituted M-type hexaferrites are excellent candidate for such magnetic material. This is the simplest type of all ferrites with a hexagonal structure. More than 90% of permanent magnets are produced all over the world based on this compound. This compound is a deep semiconductor (~ 109 Ohm*cm) at room temperature with a ferrimagnetic structure and a total magnetic moment of 20 μB in the ground state :

(1). M.A. Almessiere, et. al. Microstructural and magnetic investigation of vanadium-substituted Sr-nanohexaferrite, J. Magn. Magn. Mater. 471 (2019) 124-132. https://doi.org/10.1016/j.jmmm.2018.09.054.

This information should be mentioned in 1. Introduction section of the paper.

2.    By combining other types of polymer with ferrites and carbon–based materials the new magneto–dielectric absorbers with increased and attractive microwave absorption properties could be also fabricated :

(2). O.S. Yakovenko, et. al. Magnetic anisotropy of the graphite nanoplatelet–epoxy and MWCNT–epoxy composites with aligned barium ferrite filler, J. Mat. Sci. 52 (2017) 5345-5358. https://doi.org/10.1007/s10853-017-0776-4.

3.    Furthermore, a large spontaneous polarization and multiferroic properties at room temperature recently discovered in barium hexaferrites substituted by diamagnetic cations. Herewith the magnetoelectric characteristics of M-type hexaferrites fabricated by a modified ceramic technique are more advanced than those for the well-known room temperature BiFeO3 orthoferrite multiferroic :

(3). V.G. Kostishyn, et. al., Dual ferroic properties of hexagonal ferrite ceramics BaFe12O19 and SrFe12O19, J. Magn. Magn. Mater. 400 (2016) 327-332. https://doi.org/10.1016/j.jmmm.2015.09.011.

(4). S.V. Trukhanov, et. al. Polarization origin and iron positions in indium doped barium hexaferrites, Ceram. Int. 44 (2018) 290-300. https://doi.org/10.1016/j.ceramint.2017.09.172.

This information should be also mentioned in 1. Introduction section of the paper.

4.    In 2. Materials and Methods section of the paper, it is necessary to add information about the modernized sol-gel method of obtaining magnetic and microwave absorptionmaterials 

5.    It is well known that the 3d-metal oxides easily allow the oxygen excess and/or deficit. It is necessary to point out in 3. Results and discussions section of the paper that oxygen nonstoichiometry greatly affects the magnetic and microwave absorptionproperties of 3d-metal oxides :

(5). F. Gözüak, et. al. Synthesis and characterization of CoxZn1−xFe2O4 magnetic nanoparticles via a PEG-assisted route, J. Magn. Magn. Mater. 321 (2009) 2170-2177. https://doi.org/10.1016/j.jmmm.2009.01.008.

Oxygen excess and deficit can increase and decrease the oxidation degree of 3d-metalls. The changing of charge state of 3d-metals as a consequence of changing of oxygen content changes such magnetic parameters as spontaneous magnetic moment and Curie point. Moreover, oxygen vacancies effect on exchange interactions. Intensity of exchange interactions decreases with oxygen vacancy concentration increase. In 3d-metall oxides there is only indirect exchange. Exchange near the oxygen vacancies is negative according to Goodenough-Kanamori empirical rules. Oxygen vacancies for nanoparticles of ferrite should lead to the formation of a weak magnetic state such as spin glass. This information should be discussed in 3. Results and discussion section of the paper.

6.        In contrast to other complex oxides, i.e. perovskites and spinels, the real part of the dielectric constant for multiferroic substituted hexaferrites decreases more slowly at low frequencies and almost monotonically with diamagnetic substitution. And the real and imaginary parts of the permeability have a peak near 50 GHz, which is determined by the level of diamagnetic substitution :

(6). S.V. Trukhanov, et. al. Effect of gallium doping on electromagnetic properties of barium hexaferrite, J. Phys. Chem. Sol. 111 (2017) 142-152. https://doi.org/10.1016/j.jpcs.2017.07.014.

(7). H. Sözeri, et. al. Cr3+-substituted Ba nanohexaferrites as high-quality microwave absorber in X band, J. Alloys Compd. 779 (2019) 420-426. https://doi.org/10.1016/j.jallcom.2018.11.309.

This information should be also discussed in 3. Results and discussion section of the paper.

Author Response

List of Responses to the Reviewer’s Comments on the Manuscript

(Polymers - 479619)

Complex Permittivity and Microwave Absorption Properties of OPEFB Fiber – Polycaprolactone Composites Filled with Recycled Hematite (α – Fe2O3) Nanoparticles

Thank you for reviewing our submission and your very valuable comments. We carefully responded to all the comments and have modified the manuscript accordingly. The following are our responses to the specific comments raised.

1 – 3.

Information to be added to the Introduction section

Information mentioned in relevant portions of the Introduction on lines 44 and 68.

4.    In 2. Materials and Methods section of the paper, it is necessary to add information about the modernized sol-gel method of obtaining magnetic and microwave absorption materials 

We kindly wish to seek further clarification on this request to add the stated information to the Materials and Methods section since we have described one method of obtaining such a material from industrial waste in our research.

5 - 7.    Information to be added to the Results and discussion section

Information mentioned in relevant portions of the Results and discussion on lines 244-245.

• All the comments made have been acted upon.

Thank You.

Ebenezer Ekow Mensah

Physics Department, Faculty of Science, Universiti Putra Malaysia.

([email protected])

Reviewer 2 Report

The manuscript entitled “Complex Permittivity and Microwave Absorption Properties of OPEFB Fiber – Polycaprolactone Composites Filled with Recycled Hematite (α – Fe2O3) Nanoparticles” describes the effect of different wt.% of nanofiller derived by hematite on composite from electric point of view (Permittivity and microwave absorption).

The work needs some clarification in different points that I summarize in the follow observations:

1.      Lines 162-164: Could the authors explain why the surface area increases with the reduction of particle size?

2.      Line 179: in order to do not modify the fractured surface, cryo fractured treatments are required. Did the authors apply this treatment? If not, why?

3.      Figure 4: In these figures the scales are not clear, please modify.

4.      Line 212-216: It is not clear if the authors measure composite or only compacted nanoparticles. Please clarify.

5.      Line 224: are you sure that the permittivity decrease when the particle sizes are reduced?

6.      Figure 6 and figure 7: at low frequency the measure is unstable, is it possible to have more stable measurements? If not, please motivate the reason why the measures are unstable.

7.      It is not clear at which particle size are referred the measurements reported in figure 7.

8.      A sketch of complex permittivity instrument set up could help readers to better understand the measurements execution. I strongly suggest to add this figure.

Author Response

List of Responses to the Reviewer’s Comments on the Manuscript

(Polymers - 479619)

Complex Permittivity and Microwave Absorption Properties of OPEFB Fiber – Polycaprolactone Composites Filled with Recycled Hematite (α – Fe2O3) Nanoparticles

Thank you for reviewing our submission and your very valuable comments. We carefully responded to all the comments and have modified the manuscript accordingly. The following are our responses to the specific comments raised.

1.    Lines 162-164: Could the authors explain why the surface area increases with the reduction of particle size?

The explanation has been provided. Please check lines 169-175.

2.  Line 179: in order to do not modify the fractured surface, cryo fractured treatments are required. Did the authors apply this treatment? If not, why?

Cryo fractured treatments could not be applied to the nanocomposites due to equipment challenges. The word ‘Fractured’ was inadvertently used in the text and has been removed from the revised version. Please check line 190.

3.    Figure 4: In these figures the scales are not clear, please modify.

Figure has been modified on line 178-180

4.    Line 212-216: It is not clear if the authors measure composite or only compacted nanoparticles. Please clarify.

Authors measured compacted nanoparticles. Clarification has been made. Please check line 127-128.

5.    Line 224: are you sure that the permittivity decrease when the particle sizes are reduced?

Authors stated; “These results demonstrate that the complex permittivity of recycled α – Fe2O3 particles are significantly enhanced when the particle sizes are reduced to nanosize” on line 224.

6.    Figure 6 and figure 7: at low frequency the measure is unstable, is it possible to have more stable measurements? If not, please motivate the reason why the measures are unstable.

The reasons for the instability in the Figures referred to have been provided on lines 230-232

7.    It is not clear at which particle size are referred the measurements reported in figure 7.

The nanocomposites were fabricated using 16.2 nm recycled α – Fe2O3 nanoparticles and has been highlighted in the revised version on line 249.

8.    A sketch of complex permittivity instrument set up could help readers to better understand the measurements execution. I strongly suggest to add this figure.

The sketch has been provided and labelled Figure 2 on lines 132-134.

• All the comments made have been acted upon.

Thank You.

Ebenezer Ekow Mensah

Physics Department, Faculty of Science, Universiti Putra Malaysia.

([email protected])

Round 2

Reviewer 1 Report

Revised version can be accepted.